# Quality Research of the Beetroots (*Beta vulgaris* L., ssp. *vulgaris* var. *conditiva* Alef.) Grown in Different Farming Systems Applying Chemical and Holistic Research Methods

Aurelija Paulauskienė [1,*], Daiva Šileikienė [2], Rasa Karklelienė [3], Živilė Tarasevičienė [1] and Laima Česonienė [2]

1 Department of Plant Biology and Food Sciences, Vytautas Magnus University Agriculture Academy, Donelaičio Str. 58, 44248 Kaunas, Lithuania; zivile.taraseviciene@vdu.lt
2 Department of Environment and Ecology, Vytautas Magnus University Agriculture Academy, Donelaičio Str. 58, 44248 Kaunas, Lithuania; daiva.sileikiene@vdu.lt (D.Š.); laima.cesoniene1@vdu.lt (L.Č.)
3 Institute of Horticulture, Lithuanian Research Centre of Agriculture and Forestry, 54333 Babtai, Lithuania; rasa.karkleliene@lammc.lt
* Correspondence: aurelija.paulauskiene@vdu.lt

**Abstract:** For consumers who care about food sustainability, sustainable food seems to be at least organic, local, and seasonal food. Our research aimed to compare the differences between beetroots grown conventionally and organically using chemical and electrochemical research methods demonstrating life processes in products. Three beetroot (*Beta vulgaris* L., ssp. *vulgaris* var. *conditiva* Alef.) cultivars, 'Joniai', 'Jolie' H, and 'Grenade' H, were grown using organic and conventional farming systems. The total yield, diameter, and root length were measured after harvesting. The main chemical composition and color coordinates were determined. Holistic electrochemical research methods were applied to demonstrate the vitality of the products. It was found that the yield of conventionally grown vegetables depending on cultivars was from 4 to 19% higher than that of organically grown vegetables, and the dependence of the chemical composition on the farming system was ambiguous. Although the higher amounts of DM (13.70–15.90%), TSS (10.50–12.20%), and sugars (8.47–8.98%) were found in organically grown beetroots, higher contents of betalains (647–1408 mg kg$^{-1}$ fw) were accumulated by conventionally grown plants. The highest amount in the fresh weight of Ca (310 mg kg$^{-1}$) and Mg (470 mg kg$^{-1}$) was accumulated in conventionally grown cv 'Joniai'; only organically grown cv 'Jolie' H beetroots accumulated the highest amount of Fe (17.40 mg kg$^{-1}$). However, lower values of the electrochemical parameters rH and P showed a better quality of ecologically grown beetroot, better vitality, and better suitability for human consumption.

**Keywords:** beetroot; betalains; organic farming; conventional farming; electrochemical parameters

## 1. Introduction

Worldwide attention to food quality as well as to food sustainability is increasing [1]. Food security, seen from a global perspective, encompasses both the sustainable production of high-quality food and the reduction in food waste [2]. Agricultural transitions to sustainability may be driven as much by technological changes or institutional features, including normative and cultural differences [3]. European countries have great potential to create healthy food environments to promote healthy diets [4]. To address the challenges of globalization, the International Federation of Organic Agriculture Movements (IFOAM) formulated the principles of organic agriculture [5]. Consumer demand for healthier food and government policies regarding the environmental sustainability of agricultural processes are increasingly driving the rapid development of organic farming [6]. Nevertheless, the consumer does not fully perceive the relationship between organic products' increased nutritional and environmental values. [7].

The consumption of vegetables and fruits is essential to people's overall health. In recent years, the root vegetable *Beta vulgaris* subsp. *Vulgaris*—otherwise known as red beetroot (herein referred to as beetroot)—has attracted much attention as a health promoting functional food [8]. Beetroot is rich in bioactive compounds that may provide health benefits, particularly for disorders characterized by chronic inflammation. Recent clinical studies show that using beetroots reduces blood pressure and improves the clinical outcome of atherosclerosis, type 2 diabetes, and dementia [9–13]. Beetroots are a natural source of inorganic nitrates ($NO^{3-}$) that affect blood vessels. The positive effect of nitrates is related to their reduction in vivo to nitric oxide (NO), which performs important vascular and metabolic functions [8,10,11,13]. Beetroot is a rich source of vitamins (ascorbic acid, B complex), carotenoids, phenolic acids, and flavonoids, [14–17], as well as a source of dietary fiber and minerals (magnesium, calcium, potassium, sodium, iron, copper, phosphorus, and zinc) [14,18]. They contain a group of highly bioactive water-soluble nitrogenous pigments known as betalains [9,19,20]. More than 80% of the red beetroot pigments are composed of betacyanins, mainly betalains [21]. Beetroots are known as one of the few edible sources of betalains. Betalains are derivatives of betalamic acid and are divided into two subclasses, betacyanins (red pigments) and betaxanthines (yellow pigments). The typical color of beetroots is caused by the best-known betacyanin, betanin (and its isomer, isobetanin), which forms a major part of the total betalain content (up to 41%) [20]. Structurally, betalains are similar to anthocyanins and, within plants, perform identical functions to anhocyanins. In vitro and in vivo studies proved betalains' antimicrobial, antiviral, anti-inflammatory, and antioxidant activities [17,19,22,23]. Due to these characteristics of betalains, red beetroots were recently included in the group of vegetables with the highest antioxidative potential.

For consumers who care about food sustainability, sustainable food seems to be at least organic, local, and seasonal food [8]. Beetroots, like other root vegetables, are relatively cheap, can be produced locally worldwide, and have a long shelf life. Their cultivation does not pose problems, and the good storability properties ensure the availability of a fresh product throughout the year and do not require expensive storage equipment [24]. Although beetroots cultivation is not complicated, enough nitrogen is needed for foliage growth and phosphorus and potassium for root crops. Researchers stated that the productivity of beetroot increases with the application of nitrogen fertilizers, and the method of cultivation has a significant effect on the chemical composition of this vegetable [24,25]. A number of studies suggest that the lower yield and better nutritional quality of organically grown plants are directly related to the greater stress when they grow. When plants are more stressed due to unfavorable growing conditions when they do not use synthetic fertilizers and pesticides, their natural defences increase, and more biologically active compounds are synthesized [13,26–28].

Conventionally grown product quality is mainly understood in terms of external appearance and nutritive and sensory properties. According to Bloksma et al. [29], when consumers choose organic products, they expect them to not only be ripe and tasty but to have properties such as 'vitality' and 'coherence'. According to the Inner Quality Concept, quality is related to the life processes of growth and differentiation in plants, and since these processes occur in living organisms at the same time, they cannot be separated. Thus, this concept defines optimal food quality as a balance of life processes in plants. Researchers suggested several experimental parameters for assessing the coherence aspects of food products, and some of them were electrochemical measurements. Electrochemical research methods provide additional information about metabolism and physiological processes in plants, as they can be described as chains of electrochemical or redox reactions [29]. Electrochemical methods, such pH and rH (absolute redox potential), are fast, simple, and human and environmentally friendly, allowing for the reduction in the need for expensive and time-consuming laboratory analyses [30].

The aim of this research is to compare the differences between beetroots grown conventionally and organically using chemical and electrochemical research methods that demonstrate life processes in products.

## 2. Materials and Methods

### 2.1. Beetroots Growing Conditions

Three beetroot (*Beta vulgaris* L., ssp. *vulgaris* var. *conditiva* Alef.) cultivars were selected for the experiment: 'Joniai', 'Jolie' H, and 'Grenade' H (Table 1).

**Table 1.** Cultivar and roots description.

| Cultivar Name | Cultivar Description | Roots Description |
|---|---|---|
| 'Joniai' | Medium-early universal Lithuanian population cultivar | Roots: round or round oval shape. The flesh is red, with not clearly concentrated rings. |
| 'Jolie' H | Medium-early Dutch hybrid cultivar | Roots: uniform round shape. The flesh is bright red, without ring inserts. |
| 'Grenade' H | Medium-early French hybrid cultivar | Roots: round to highly round shape. The flesh is dark red. |

The experiment was carried out in 2020 on Calcic Endogleyic Luvisol (LV-gl-n-cc) light loam [31], located at the Institute of Horticulture of the Lithuanian Research Center for Agriculture and Forestry (55°04′55.7″ N 23°47′54.5″ E). Beetroots were grown using organic and conventional farming systems. The main properties of the soil plough layer and used fertilizers are presented in Table 2.

**Table 2.** Soil properties and used fertilizers in the experiment.

| | Conventional Growing System | Organic Growing System |
|---|---|---|
| Soil $pH_{KCl}$ | 7.2 | 6.9 |
| Humus % | 3.5 | 5.5 |
| Available phosphorus ($P_2O_5$), mg kg$^{-1}$ | 280 | 340 |
| Available potassium ($K_2O$), mg kg$^{-1}$ | 213 | 265 |
| Total nitrogen ($N^-NO_3 + N^-NH_4$), mg kg$^{-1}$ | 26.3 | 31.4 |
| Electrical conductivity (EC), mS m$^{-1}$ | 0.72 | 0.49 |
| Fertilizers | 'Cropcare 11-11-21 and microelements'—500 kg ha$^{-1}$ | 'Ecoplant'—400 kg ha$^{-1}$, 'Activit'—100 kg ha$^{-1}$ |

Compositions of organic fertilizers (100% natural product):

The 'Ecoplant' (ash from sunflower husk), (%): Total phosphorus ($P_2O_5$)—4.0; Total potassium ($K_2O$)—28.0; Total organic carbon and organic matter—36.0; Microelements (Ca, Mg, S, Mn, Zn, and other)—32.0; pH—10.5.

The 'Activit' (made from chicken manure), (%): Organic matter—62.0; Total nitrogen (N)—3.6; Total phosphorus ($P_2O_5$)—2.8; Total potassium ($K_2O$)—2.2; Microelements (Ca, Mg, Mn, Zn, and other)—29.4; pH—6.4.

Beetroots were sown on 5 May in two rows, with a spacing between rows of 70 cm. Fertilizers were applied before sowing beetroots. Until the beginning of August, the beetroots were additionally watered, the space was loosened between rows twice, and they were weeded three times. The beetroots were harvested on 29 September. The experiments were carried out in three replicates.

After beetroots harvesting, the total yield (t ha$^{-1}$) was assessed, and the diameter (cm) and root length (cm) were measured.

### 2.2. Meteorological Conditions

The average air temperature in May was 1.5 °C lower compared to the long-term mean value, and the rainfall exceeded 25.7 mm (Figure 1). The weather was favorable, and the

beetroots sprouted completely on 25 May. In June, the temperature was higher, at 3.3 °C, and the rainfall was almost twice as high as the long-term averages. Although July was colder and drier, the growth of beetroots was not significantly affected. Beetroot growth intensified in August, when rainfall was sufficient. August and September were favorable for the ripening of roots, and in mid-September, the beetroots reached the shape typical of the cultivar.

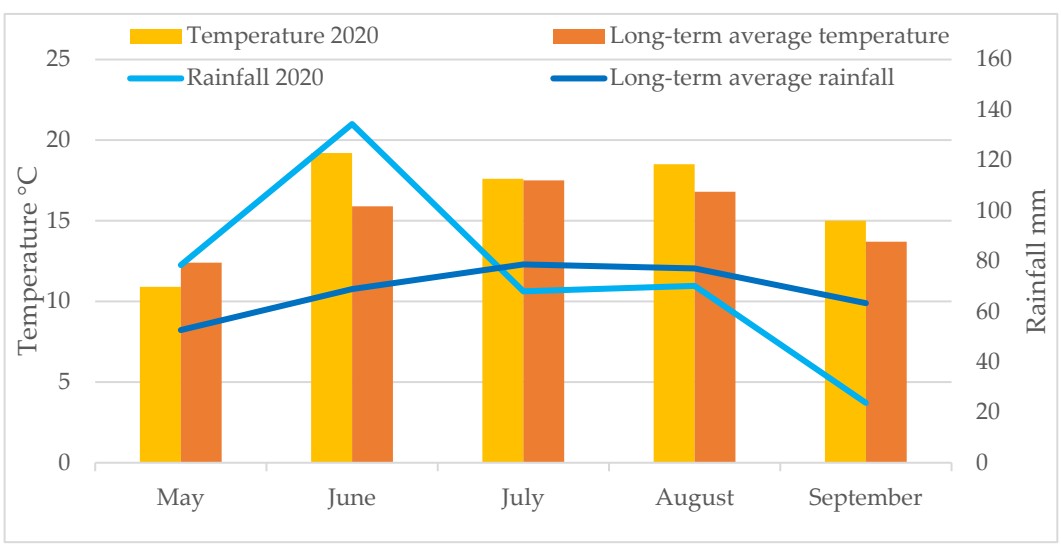

**Figure 1.** Average temperature and rainfall sum during the beetroots vegetation period (Babtai, 2020).

### 2.3. Laboratory Samples Preparation

For the determination of the chemical composition, physical properties, and holistic tests of the beetroots of each cultivar, a laboratory sample weighing at least 1 kg was prepared from all three replicates. No less than 2 kg of the beetroots of each cultivar was taken to the laboratory for each replicate (total 6 kg) to form a laboratory sample. Before analysis, the laboratory samples were stored in a refrigerator at a temperature of 5 °C.

### 2.4. Chemical Analysis of Soil

The soil $pH_{KCl}$ was measured in 1 N KCl extraction by the potentiometric method [32]; available phosphorus ($P_2O_5$) and available potassium ($K_2O$) were measured by the Egner–Riehm–Domingo (A–L) method [33].

### 2.5. Determination of the Chemical Composition and Physical Properties of Beetroots

Chemical analyses were performed in three replications. Analytical grade chemicals used in this study were purchased from Labochema LT (Vilnius, Lithuania).

The dry matter (DM) content was assessed by drying the samples of shredded beetroots to a constant mass at 105 °C [34]; the soluble solids concentration (TSS) was measured in the juice pressed from homogenized beetroots mass by using a digital refractometer PAL-1 (Atago, Japan) at 20 °C [34].

Briefly, 2.5 g of the homogenized beetroots sample was added to 200 mL 40% ethanol and mixed in the shaker for an hour. Then, 5 mL Carez solution I was added and stirred for approximately 30 s. After adding 1 mL Carez solution II, it was stirred for one minute. After that, the volume was made up to 250 mL with 40% ethanol, mixed, and filtered. Then, 200 mL filtrate was evaporated to approximately half volume to eliminate most of the ethanol. The residue was poured into a 200 mL volumetric flask and made up to a volume with water, mixed, and filtered. Then, the solution was used to determine the amount of sugars before and after inversion by the Luff–Schoorl method [35].

For the betalains analysis, a 1 g homogenized beetroots sample was mixed with 50 mL of 50% ethanol solution. The prepared mixture was stirred for approximately 10–15 s, and

the homogenate was centrifuged at 6000 rpm for 10 min. The supernatant was collected, and the same procedure was repeated two more times to ensure maximum extraction. The supernatant was used for betalains determination by a two-ray UVS2800 spectrophotometer (Labomed Inc., Los Angeles, CA, USA), as described by Ravichandran et al. [36].

The content of mineral elements was determined. The calcium (Ca) content was established according to Commission Directive 71/250/EEC [35]. The ash was treated with hydrochloric acid, and the calcium precipitated as calcium oxalate. The precipitate was dissolved in sulphuric acid, and the formed oxalic acid was titrated with potassium permanganate solution. Magnesium (Mg) and iron (Fe) contents were identified by atomic absorption spectrophotometry [37]. The results were recalculated and presented in fresh weight (fw).

The color coordinates of brightness, redness, and yellowness—L*, a*, and b*, according to the CIELAB scale—were determined by a color analyzer ColorFlex EZ (Hunter Addociates Laboratory, Inc., Reston, VA, USA).

### 2.6. Analysis of Electrochemical Parameters

Electrochemical parameters were measured in homogenized beetroots mass. The pH and redox potential (Eh) were measured by a 781 pH/Ion Meter (Metrohm AG, Herisau, Switzerland), and electrical conductivity (electrical conductivity is the reciprocal of electrical resistivity) was measured by a laboratory conductivity meter inoLab® Cond 7310 (Xylem Inc., WTW, Weilheim, Germany). The P value as a combined parameter of the three mentioned parameters was calculated according to the following formula:

$$P = [29.07 \, (rH - 2 \, pH)]^2 \cdot rHo^{-1} \, (\mu W), \tag{1}$$

where rH—absolute redox potential; pH—the hydrogen-ion activity; rHo—recalculated specific electrical conductivity ($\mu S \, cm^{-1}$) [38].

rH—absolute redox potential was calculated according to the formula [38]:

$$rH = [(Eh + 200) \cdot 30^{-1}] + 2 \, pH, \tag{2}$$

### 2.7. Statistical Analysis

Data analysis was carried out with STATISTICA version 7 software (TIBCO Software, Palo Alto, CA, USA). The results were analyzed by using a factorial analysis of variance (ANOVA). The arithmetical means and standard deviations of the experimental data were calculated. Fisher's Least-Significant-Difference test (LSD) was applied to the experimental results to assess significant differences between mean values at the significance level of $p < 0.05$. The principal component analysis (PCA) was performed to evaluate the relationships between the different farming systems and the chemical composition, physical properties, and electrochemical parameters of beetroots with XLSTAT software version 2019.3.02 (Addinsoft, Paris, France).

## 3. Results and Discussion

### 3.1. Beetroots Yield Characteristic

Red beetroots are one of the main vegetables grown in Lithuania. Beetroot is not a very demanding vegetable, and it can be grown in different soils. These plants, with a long growing season, take most of the nutrients from the fertilizers applied before sowing. However, it is natural that nutrients begin to run out during the vegetation. During the growing, it is recommended to fertilize the beets through the leaves; additionally, no additional fertilization was used in our study. The results obtained by researchers confirmed the statistically significant effect of the cultivar on the yield of beetroots [39]. The beetroots yield can be affected by the different growing systems, irrigation and soil salinity, fertilization strategy, or meteorological conditions during the vegetation period [39]. The yield of beetroot may vary from 19.86–25.70 [40] or 20.74–41.91 [39] to 54.80–69.43 t ha$^{-1}$ [24]. Szopińska and Gawęda stated that when analyzing the total yield from the three years of

cultivation, no significant influence of the cultivation method on the yield was noted, but the obtained results showed a strong impact of weather conditions during the vegetation period on the beetroots yield [40]. The yield of beetroots in our experiment was several times higher compared with the results obtained by the two mentioned studies and ranged from 52.91 to 66.70 t ha$^{-1}$ (Table 3). The higher yield could be influenced by the favorable meteorological conditions during the vegetation period; since June, the temperature was higher compared to the long-term average, and there was a sufficient amount of precipitation at the beginning of seed germination and growth. Higher results were obtained for beetroots grown using a conventional farming system. The conventionally grown cv. 'Joniai' yield was about 4% higher than that of organically grown ones, and those of 'Jolie' H and 'Grenade' H were almost 19% and 14% higher than those of organically grown ones, respectively. The highest yield was that of conventionally grown 'Grenade' beetroots. The obtained results are also confirmed by the experiments conducted in the Lithuanian Research Center of Agriculture and Forestry under the plant breeding program, where the total yield of conventionally grown beetroot was even 25% higher than that of organically grown beetroot [41]. In general, it is believed that yields in organic agriculture are 10–40% lower when compared to the conventional one [26,42]. However, other researchers obtained opposite yield results, i.e., the total yield of organically grown beetroots was about 23% higher than that of conventionally grown beetroots and 10% higher than that of integrated grown beetroots [40].

**Table 3.** Beetroots' length, diameter, and total yield.

| Farming Systems/Cultivars | 'Joniai' | 'Jolie' H | 'Grenade' H |
|---|---|---|---|
| Root length, cm | | | |
| Conventional | 9.3 ± 0.20 b | 9.8 ± 0.30 a | 9.6 ± 0.41 ab |
| Organic | 8.4 ± 0.30 c | 8.7 ± 0.10 c | 8.7 ± 0.20 c |
| Root diameter, cm | | | |
| Conventional | 7.7 ± 0.25 a | 7.5 ± 0.18 a | 7.8 ± 0.16 a |
| Organic | 7.6 ± 0.26 a | 7.5 ± 0.17 a | 7.6 ± 0.28 a |
| Total yield, t ha$^{-1}$ | | | |
| Conventional | 63.07 ± 1.09 b | 63.13 ± 1.68 b | 66.70 ± 0.88 a |
| Organic | 60.53 ± 1.56 c | 52.91 ± 1.59 d | 58.73 ± 1.39 c |

Significant differences ($p < 0.05$) for each parameter between the cultivars and the farming systems are marked by different letters; for each measured parameter, the general mean ± SD is presented.

Measuring the length of the roots, the conventionally grown roots of our experiment were 1.1-fold extended compared to organically grown roots (Table 3). Similar results were obtained by Szopińska and Gawęda [40]. 'Jolie' conventional roots were the longest (9.8 cm). The farming system did not significantly affect the diameter of the beetroots in our experiment and in those of other researchers [40].

*3.2. Beetroots' Chemical Composition and Physical and Electrochemical Parameters*

Beetroots' DM (dry matter) content ranged from 12.60 to 16.30%, their TSS (total soluble solids) ranged from 10.00 to 12.20%, and their sugars ranged from 8.12 to 8.98% (Figure 2). The research data showed that the higher amounts of DM, TSS, and sugars were in organically grown beetroots. Fjelkner-Modig et al. [42] stated that organically grown crops were found to accumulate 2.9% higher DM content than the conventionally grown ones, although for the different crops, contradictory results can be noted. Conventionally grown 'Joniai' accumulated about 1-fold more DM, but this difference was insignificant. Regardless of the farming system, 'Joniai' beetroots accumulated the highest amounts of DM, TSS, and sugars. Several authors reported a higher content of sugars in organic vegetables, including beetroots, tomatoes, carrots, potatoes, and others, as well as in fruits [43,44].

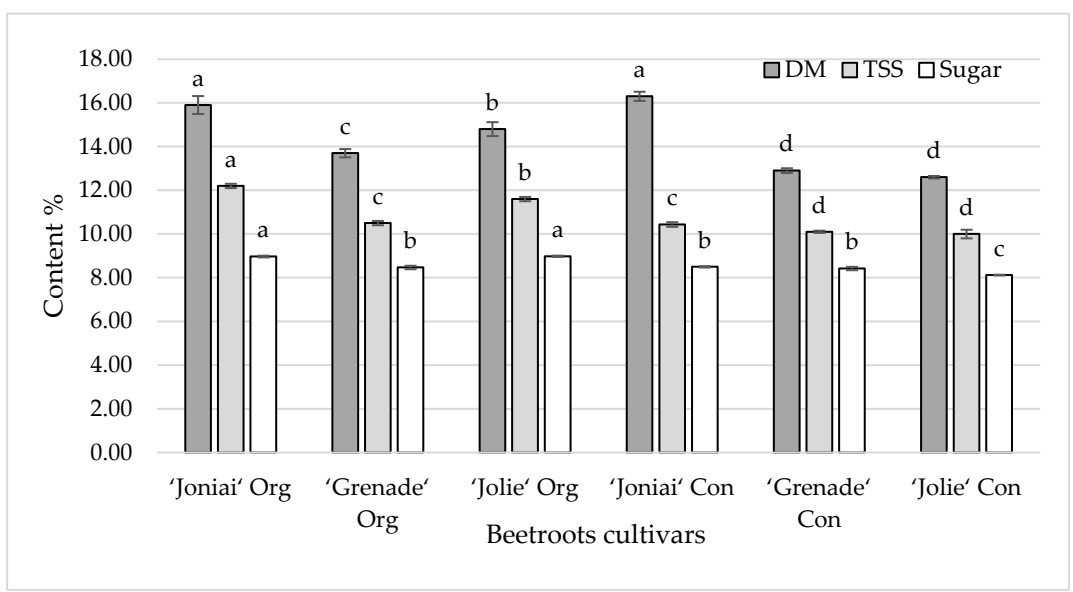

**Figure 2.** Dry matter (DM), total soluble solids (TSS), and sugars % content of conventionally (Con) and organically (Org) grown beetroots. Significant differences ($p < 0.05$) for each parameter between the cultivars and the farming systems are marked by different letters.

Many studies show that the content of betalains can vary greatly among cultivars as well as growing conditions: 650 and 800 $\mu$g g$^{-1}$ fresh weight [45], 120 mg 100 g$^{-1}$ dry weight, or 700 and 1300 mg L$^{-1}$ in beetroots fresh juice [46]. In our study, the total betalains content varied from 647 to 1408 mg kg$^{-1}$ fw (Figure 3). Significant differences were found between the amounts of betalains in different cultivation systems and in all cultivars of beetroot. Conventionally grown cv 'Joniai' and 'Jolie' beetroots accumulated 1.0 to 1.4-fold higher amounts of betalains compared with organically grown ones. The cv 'Grenade' betalain content was the opposite, depending on the cultivation system; organically grown 'Grenade' beetroots accumulated a 1.1-fold higher content of these compounds. The significantly highest content of betalains was found in cv 'Joniai' beetroots, regardless of the farming system.

Szopińska and Gawęda's [40] research results showed higher amounts of betalains in conventionally grown plants, while Kosson et al. [25] indicated the higher content of betanine in organic beetroots and confirmed that the content depends on the cultivar, planting time, fertilization, and climatic conditions. A positive correlation between the increasing intensity of light and the content of betalains has been reported [45]. The intensity of light increases the activity of tyrosinase catalyses and the formation of betalain precursors, and the synthesis of betalain in beetroots intensifies [23,47]. Other scientific studies showed that the number of betalains depends on the plant's vegetation continuance; the roots of longer vegetation accumulated higher amounts of these compounds [48].

The color of beetroots is influenced by the content of pigments, especially betalains. Betalains are of interest to researchers in various fields and the food industry because of their use as naturally occurring colorants and as compounds with health-preventive properties. The color analysis of the investigated beetroots showed that organically grown ones were darker compared to conventionally grown ones (Table 4). Conventionally and organically grown 'Grenade' beetroots' flesh was the darkest. A higher red color intensity was found for conventionally grown 'Joniai' and 'Jolie' and for organically grown 'Grenade'. The most intense red color was found for conventionally grown 'Joniai' beetroots. Organically grown beetroots were more intensely yellow than conventionally grown ones. The most intense yellow color was found for organically grown 'Joniai' beetroots.

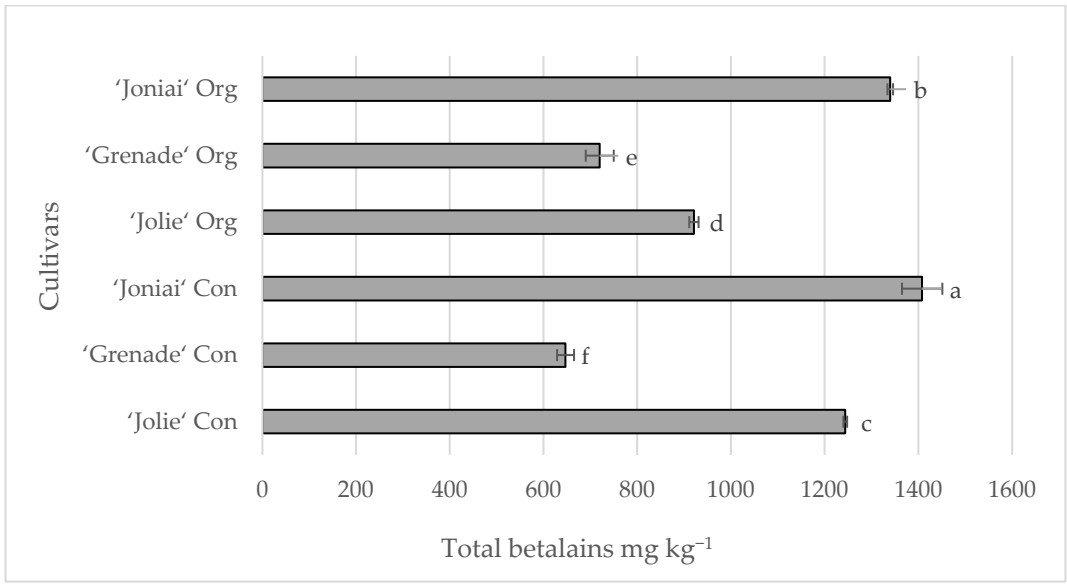

**Figure 3.** Total betalains content mg kg$^{-1}$ (fw) of conventionally (Con) and organically (Org) grown beetroots. Significant differences ($p < 0.05$) for each parameter between the cultivars and the farming systems are marked by different letters.

**Table 4.** Color parameters of conventionally and organically grown beetroots.

| Farming Systems/Cultivars | 'Joniai' | 'Jolie' H | 'Grenade' H |
|---|---|---|---|
| | | L* | |
| Conventional | 6.59 ± 0.16 b | 7.21 ± 0.18 a | 4.53 ± 0.13 e |
| Organic | 5.66 ± 0.09 c | 5.30 ± 0.09 d | 4.02 ± 0.05 f |
| | | a* | |
| Conventional | 19.38 ± 0.15 a | 17.48 ± 0.16 c | 15.79 ± 0.29 e |
| Organic | 18.91 ± 0.16 b | 16.94 ± 0.20 d | 16.04 ± 0.05 e |
| | | b* | |
| Conventional | 3.75 ± 0.19 b | 3.13 ± 0.13 d | 3.07 ± 0.18 d |
| Organic | 4.27 ± 0.14 a | 3.37 ± 0.05 c | 3.19 ± 0.04 cd |

Significant differences ($p < 0.05$) for each parameter between the cultivars and the farming systems are marked by different letters; for each measured parameter, the general mean ± SD is presented.

The number of mineral substances is influenced by the plant variety, the soil properties, the fertilizers used, the meteorological conditions during the growing season, and the maturity of the vegetables [49,50]. The genetic characteristics of beetroot varieties have a significant influence on their mineral composition, regardless of the cultivation method [50]. Minerals are vital components that perform many different functions to allow for the optimal behavior of plants as well as the human body. Beetroots accumulate large amounts of calcium (Ca) (24–28 mg 100g$^{-1}$ fw), sodium (Na), potassium (K) (279–300 mg 100 g$^{-1}$ fw), magnesium (Mg) (30–33 mg 100g$^{-1}$ fw), and iron (Fe) (1.23–1.51 mg 100g$^{-1}$ fw) [49,50]. Wruss et al. [49] found significant differences in the concentration of mineral compounds in juice prepared from different beetroot varieties.

The investigated beetroots accumulated Ca from 190 to 310 mg kg$^{-1}$, Mg from 280 to 470 mg kg$^{-1}$, and Fe from 11.90 to 17.40 mg kg$^{-1}$ fw (Table 5). The beetroots from our research were valuable vegetables because of their mineral content. According to Regulation (EU) No 1169/2011 [51], the daily consumption of 100 g of beetroot can ensure 2–4% of Ca, 8–13% of Mg, and 9–12% of Fe, a daily requirement.

**Table 5.** Content of mineral elements (mg kg$^{-1}$ (fw)) in conventionally and organically grown beetroots.

| Farming Systems/Cultivars | 'Joniai' | 'Jolie' H | 'Grenade' H |
|---|---|---|---|
| | | Ca | |
| Conventional | 310.00 ± 3.00 a | 220.20 ± 2.00 d | 190.00 ± 1.00 e |
| Organic | 230.00 ± 3.00 c | 250.00 ± 3.05 b | 196.67 ± 1.63 e |
| | | Mg | |
| Conventional | 470.00 ± 3.00 a | 280.02 ± 1.64 f | 410.05 ± 1.03 d |
| Organic | 420.11 ± 1.01 c | 370.31 ± 0.60 e | 460.00 ± 0.53 b |
| | | Fe | |
| Conventional | 15.00 ± 0.04 c | 11.90 ± 0.04 f | 12.30 ± 0.15 e |
| Organic | 15.70 ± 0.10 b | 17.40 ± 0.10 a | 13.90 ± 0.10 d |

Significant differences ($p < 0.05$) for each parameter between the cultivars and the farming systems are marked by different letters; for each measured parameter, the general mean ± SD is presented.

Domagała-Świątkiewicz & Gąstoł's [52] research in Poland has shown that the content of mineral components identified in the beetroots cultivated using organic or conventional methods can be significantly influenced by the method of cultivation. The difference in the soil K and Mg quantity caused by the farming system was found to be significant; the soil K and Mg concentration was higher for conventional fields than for the organic ones. Based on research with various fruits and vegetables, organic vegetable plantation soils had higher Ca concentrations [52]. The statistical analysis of data from research conducted with beetroot in Croatia showed variations in the content of minerals depending on the initial soil nutrient content and weather conditions (total precipitation and mean daily temperature), as well as on differences in the nutrient-holding capacity of the soil [50]. According to this research, organic fertilization had a positive effect on beetroot P content but had no significant effect on microelement uptake. Some other studies reported significantly increased availability of Zn, Fe, and Mn [50]. Furthermore, the interaction of soil macronutrients and micronutrients affects micronutrient uptake. The soil K content had a big role in decreasing the uptake of the macroelements Ca and Mg in beetroots, as K has an antagonistic relationship with those elements [50]. All these results suggest that the best results could be expected from the combined fertilization of organic and mineral fertilizers and additional microelements, especially by foliar fertilization in extreme climatic conditions [50,52].

In our research, the organic farming system had a positive effect on the number of mineral elements in two beetroot cultivars (Table 5). The higher amounts of Ca, Mg, and Fe were found in organically grown 'Jolie' and 'Grenade' beetroots. Conventionally grown cv 'Joniai' accumulated a higher amount of Ca and Mg compared with organically grown cv 'Joniai' and the highest amount of Ca and Mg among all three cultivars, while cv 'Jolie' showed the highest amount of Fe.

Beetroot pH is an important indicator influencing the stability of betalains and color. Betalains are known to be stable at pH values between 3.0 and 7.0. The pH value of investigated beetroots differed depending on the cultivar and growing method from 6.09 to 6.16 (Table 6). Viskelis et al. [53], after analyzing 11 cultivars of beetroot grown according to the integrated plant cultivation technology, indicated that the pH values were 5.70–5.78. In our research, a higher pH value was established for organically grown ones, which differed significantly from those of the same cultivars grown conventionally. Conventionally grown beetroots of all cultivars differed significantly in terms of pH values, while the organically grown cv 'Jolie' differed significantly from the other cultivars.

**Table 6.** Electrochemical parameters of conventionally and organically grown beetroots.

| Farming Systems/Cultivars | 'Joniai' | 'Jolie' H | 'Grenade' H |
|---|---|---|---|
| pH | | | |
| Conventional | 6.14 ± 0.01 bc | 6.09 ± 0.01 e | 6.11 ± 0.01 d |
| Organic | 6.16 ± 0.01 a | 6.13 ± 0.01 c | 6.15 ± 0.01 ab |
| rH | | | |
| Conventional | 16.95 ± 0.11 c | 18.22 ± 0.09 a | 17.63 ± 0.14 b |
| Organic | 14.97 ± 0.06 f | 16.03 ± 0.04 d | 15.80 ± 0.14 e |
| *p* value μW | | | |
| Conventional | 5.86 ± 0.03 c | 6.29 ± 0.05 a | 6.07 ± 0.06 b |
| Organic | 3.19 ± 0.02 f | 4.50 ± 0.05 d | 3.95 ± 0.07 e |

Significant differences ($p < 0.05$) for each parameter between the cultivars and the farming systems are marked by different letters; for each measured parameter, the general mean ± SD is presented.

pH, redox potential, specific electrical conductivity, and product P value are indicators of holistic research and the main indicators that determine the energy value of the product, which affects the assessment of the suitability of the product for the human organism [54]. The redox potential reflects the gradient of electrons that life processes utilize for their cellular work [55]. Garban [56] stated that when rH < 28.3, the systems are reducing and can release electrons to other systems with lower rH values. A lower redox potential enables the plant cells to use free enthalpy for their activity, and such plants are more suitable for human organisms [55]. The investigated beetroots' absolute redox potential (rH) varied from 14.97 to 18.22 (Table 6). The conventionally grown beetroots' rH was higher by about 1.1-fold than that of the organically grown ones. The highest rH values were determined for cv 'Jolie' beetroots regardless of the growing method. The lower rH values of organic beetroots indicate their higher antioxidant potential, which is beneficial for human consumption [36].

P value calculation is used as an integrated method of product quality assessment. The P value defines the vitality of the organism and energy distribution tendencies. Higher P values can be interpreted as more openness entropy of the system, while lower values are a sign of ordering or coherence [29,55]. Ergun [54] claims that according to the bio-electric Vincent method, a lower rH and P value are specified as a better product quality. This method has been applied and proved to be valuable in determining the quality of fruits and vegetables receiving organic-based fertilizers or grown organically to compare with conventionally grown ones, including apples, oranges, strawberries, plums, peaches, grapes, pumpkins, carrots, tomatoes, and mushrooms [54].

P values of 3.19 to 6.19 μW were determined for beetroots. The P value of organically grown beetroots of all cultivars, like the rH value, was lower than that of the conventionally grown ones.

Principal component analysis (PCA) was performed to evaluate the relationships between the different farming systems used and the chemical composition, physical properties, and electrochemical parameters of beetroots (Figure 4).

The first two components (PCs) were associated with eigenvalues higher than one and explained 76.75% and 19.50% of the total variance in the data. As can be seen in Figure 4, a separation of the samples clearly occurs, based on the beetroots growing and some chemical content parameters, as well as physical properties. All of the organically grown beetroots were distributed at positive PC1 values, while all conventionally grown beetroots were distributed all over the map. The organically grown beetroots were closely related to the total soluble solids, sugars, pH, Mg, and Fe.

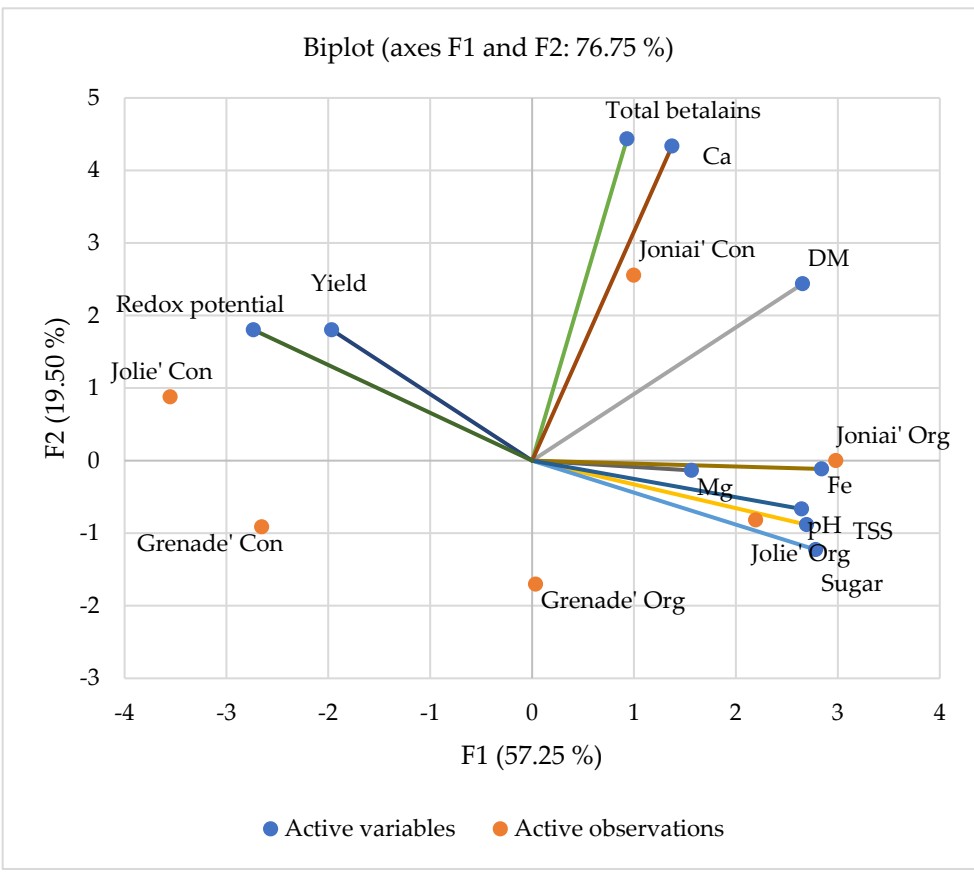

**Figure 4.** Principal component analysis (PCA) results.

## 4. Conclusions

The study of the organically and conventionally grown beetroots was a one-year study and showed that the dependence of the chemical composition on the farming system is ambiguous; therefore, further and more detailed research is required.

The study results showed that the yield of conventionally grown vegetables, depending on cultivars, was from 4 to 19% higher than that of organically grown ones. The highest yield was that of conventionally grown 'Grenade' beetroots.

The research data showed that the higher average amounts of DM, TSS, and sugars were in organically grown cv 'Joniai' and 'Jolie' beetroots, but conventionally grown cv 'Joniai' and 'Jolie' accumulated higher amounts of betalains compared with organically grown counterparts. Higher contents of Ca, Mg, and Fe were found in organically grown cv 'Jolie' H and 'Grenade' H beetroots compared to conventional ones. However, the highest amount of Ca and Mg accumulated in conventionally grown cv 'Joniai', and only organically grown cv 'Jolie' H beetroots accumulated the highest amount of Fe.

An analysis of electrochemical parameters was used to define the vitality of the vegetables and their energy distribution tendencies. Lower rH and P values were established for organically grown beetroots, which specified a better product quality.

Comparing the three studied beetroot cultivars indicated that cv 'Joniai', regardless of the cultivation method, accumulated the largest amounts of DM, TSS, sugars, Ca, and Mg. The electrochemical parameters, rH, and P values of cv 'Joniai' were also the lowest.

**Author Contributions:** Conceptualization, A.P. and D.Š.; methodology, A.P. and D.Š.; software, Ž.T.; validation, A.P., D.Š. and Ž.T.; investigation, R.K. and D.Š.; data curation, L.Č.; writing—original draft preparation, A.P. and D.Š.; writing—review and editing, Ž.T.; visualization, A.P. All authors have read and agreed to the published version of the manuscript.

**Funding:** This research received no external funding.

**Data Availability Statement:** Not applicable.

**Conflicts of Interest:** The authors declare no conflict of interest.

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
