# Peer review of "Quality Research of the Beetroots (Beta vulgaris L., ssp. vulgaris var. conditiva Alef.) Grown in Different Farming Systems Applying Chemical and Holistic Research Methods"

_sustainability, doi:10.3390/su15097102_

Round 1
Reviewer 1 Report
I have reviewed the manuscript entitled “Quality research of the beetroots (Beta vulgaris L., ssp. vulgaris var. conditiva Alef.) grown in different farming systems applying chemical and holistic research methods”. Manuscripts are well written. With addition of following comments recommending this manuscript for publication with a minor revision.
1. Rewrite the abstract part with numerical data.
2. Line 42-43: Not Clear. Rewrite with latest citation.
3. Line 59: Write more about betalains.
4. Line 143: What type of analystical grades, Plz explain. From where you procured?
5. Line 153: Why the solution was cooled?
6. Rewrite the conclusion part and include numerical data.
7. Have a good eye on grammatical errors and vocabulary as it was not up to the standard.
Author Response
Thank you for your comments. All comments have been considered and corrected.
- The abstract was rewritten with numerical data.
- Line 42-43: the sentence was rewritten: “Nevertheless, the consumer does not fully perceive the relationship between organic products' increased nutritional and environmental values”
- Line:59. More about betalains: “Betalains are derivatives of betalamic acid and are divided into two subclasses, betacyanins (red pigments) and betaxanthines (yellow pigments). For the typical colour of beetroots is responsible the best-known betacyanin betanin (and its isomer isobetanin) which form a major part of the total betalain content (up to 41%).”
- Line 143: “Analytical grade chemicals used in this study were purchased from Labochema LT (Vilnius, Lithuania).”
- Line 153: It was a mistake, and the word has been deleted.
- The conclusion part was rewritten; numerical data were included in the Abstract section.
7. The text has been reviewed.
Reviewer 2 Report
The author found that the yield of conventionally grown vegetables depending on cultivars was from 4 to 19% higher than organically grown and the dependence of the chemical composition on the farming system was ambiguous. Although the higher amounts of DM, TSS and sugars were found in organically grown beetroots, higher numbers of betalains were accumulated by conventionally grown plants. The highest amount of Ca and Mg accumulated conventionally grown cv ‘Joniai’. However, lower values of electrochemical parameters rH and P showed better quality of ecologically grown beetroot, their vitality and better suitability for the human’s consumption. Please add more relevant research from previous studies,because there is limited research on previous studies in introduction.
Author Response
Thank you for your comments. The authors have added a more extensive explanation and sources that provide results of similar studies.
“The redox potential reflects the gradient of electrons that life processes utilize for their cellular work [56]. Garban [57] stated that when rH < 28,3 the systems are reducing and can release electrons to other systems with lower rH. Lower redox potential enables the plant cells can use free enthalpy for their activity and such plants are more suitable for human organisms [56].”
“P value defines the vitality of the organism and energy distribution tendencies. Higher P values can be interpreted as more openness entropy of the system, while lower values are a sign of ordering or coherence [29,56].”
Reviewer 3 Report
The manuscript is well done. It is easy to understand and is an important issue in terms of sustainable agriculture. The article can be published as it is.
Author Response
Thank you for your comments.
Reviewer 4 Report
Overall, this paper is writing up in a professional scientific way. It will be accepted after minor revision. The authors have a sound knowledge of theoretical science and choose a very hot topic that has significance.
Remarks
1. Section of literature review can be improved.
2.Line218-219. According to table 3, the highest yield was achieved by conventionally grow Grenades, not Jolie. I recommend checking he results.
3.Mistakes. Line 162, 263 ,, The number,,
4.In figure 1 check F1 values.
5.Should have a standalone for limitations and future research
6.The English should be improved
Author Response
Thank you for your comments. All comments have been considered and corrected.
- The section of the literature review was supplemented.
- Line 218-219. The mistake has been corrected.
- Mistakes. Line 162, 263. The mistakes have been corrected.
- Figure 1 presented meteorological data.
- The conclusion part was rewritten with observations about future research.
- The text has been reviewed.
Reviewer 5 Report
The most important drawback of this research is that it is a one-year study because the results of one cropping season are usually insufficient to conclude comparing fertilizer systems. In addition, some flaws were detected in the text of the manuscript as follows:
1) The word "number" is used many times in the manuscript (lines 24, 162, 255, 257, 263, 282, 313, 314, 370, and 372). It seems that the word "number" is not used correctly in the manuscript. Because the parameters defined by "number" (betalains, mineral elements) are uncountable, besides their measurement unit (mg/kg) also shows this. It is better to use other words such as "amount" or "content" instead.
2) The chemical composition of the organic fertilizers has not been presented.
3) The results of soil analysis are very brief because important items such as total N, mineral N, EC (electrical conductivity), C/N ratio, and TNV (total neutralizing value) are not reported in it.
4) In section 3.1, what is the reason for the higher total yield in the conventional system compared to the organic system? Discussing is needed.
5) In line 219 (section 3.1), it has been noted that the highest yield was recorded by 'Jolie', while according to table 3, the highest yield was obtained by conventionally grown 'Grenade', not Jolie!
6) The reported results in lines 313-316 (highlighted in the reviewed file) are not conformed to the results in table 5. In addition, no explanation has been provided to interpret the difference between the two systems in terms of the amount of minerals.
7) The conclusion section of the paper is only presented as a summary of the results, while the conclusion should include the most important results in addition to the practical aspects of the research and provide strategies and suggestions for future research. It is also possible to point out the problems, limitations, and ambiguous aspects of the research in the conclusion.

Author Response
Thank you for your comments. All comments have been considered and corrected.
The authors agree with your observation that this one-year study requires further studies.
- Mistakes have been corrected.
- The composition of organic fertilizers was included in the section “Materials and Methods”.
- Soil analysis data were supplemented with total nitrogen content and electrical conductivity.
- The discussion about the beetroots yield was extended: “Results obtained by researchers confirmed the statistically significant effect of cultivar on the yield of beetroots [39]. Beetroots yield can be affected by the different growing systems, irrigation and soil salinity, fertilisation strategy, or meteorological conditions during the vegetation period [39].”
“Szopińska and Gawęda stated that when analysing the total yield from the three years of cultivation, no significant influence of the cultivation method on the yield was noted, but obtained results showed a strong impact of weather conditions during the vegetation period on the beetroots yield.”
- The mistake has been corrected.
- The discussion on the results of mineral elements in different beetroot cultivars has been clarified. Explanation of the difference between the two systems in terms of the amount of minerals added.
7. The conclusion section has been corrected.
Round 2
Reviewer 1 Report
The authors have thoroughly addressed all comments and suggestions made by reviewers. Now the quality of the manuscript is significantly improved. Thus, the current form of the manuscript can be accepted for publication.
Reviewer 5 Report
The corrections have been done.